# Prospective Interpersonal and Intrapersonal Predictors of Initiation and Cessation of Non-Suicidal Self-Injury among Chinese Adolescents

**DOI:** 10.3390/ijerph17249454

**Published:** 2020-12-17

**Authors:** Hui Wang, Quanquan Wang, Xia Liu, Yemiao Gao, Zixun Chen

**Affiliations:** Institute of Developmental Psychology, Beijing Normal University, Beijing 100875, China; xiaohui.wang@mail.bnu.edu.cn (H.W.); mira@bnu.edu.cn (Q.W.); 201821061064@mail.bnu.edu.cn (Y.G.); 201821061060@mail.bnu.edu.cn (Z.C.)

**Keywords:** non-suicidal self-injury, interpersonal factors, intrapersonal factors, initiation, cessation, adolescents

## Abstract

(1) *Purpose*: Non-suicidal self-injury (NSSI) possibly emerges as well as remits in adolescence. To explore the development and transition of NSSI, this study examined the association between a wide range of interpersonal and intrapersonal predictors of NSSI initiation and cessation. (2) *Methods*: Chinese adolescents (*N* = 913) completed self-reported surveys at baseline and at a six-month follow-up. The sample included 625 adolescents who reported no NSSI and 288 adolescents who reported engagement in NSSI at baseline. (3) *Results*: Among the adolescents without NSSI at baseline, 24.3% engaged in NSSI at follow-up (NSSI initiation group). Among the adolescents with NSSI at baseline, 33.3% reported no NSSI at follow-up (NSSI cessation group). Loneliness, beliefs about adversity, problem behavior, and prosocial behavior were the significant factors in predicting subsequent NSSI initiation. None of the potential predicting factors were associated with subsequent NSSI cessation. (4) *Conclusions*: These results indicate the importance of intrapersonal factors in Chinese culture, which could be used to identify at-risk adolescents and to design interventions.

## 1. Introduction

Non-suicidal self-injury (NSSI) is the intentional damage to one’s body tissue (e.g., scraping the skin and self-battery) without suicidal intent [1]. It is well documented that NSSI is a potent risk factor for suicidal thoughts and behavior [2]. As a developmental period characterized by rapid biological and psychological change, adolescence is a high-risk period for NSSI engagement [3], even among nonclinical adolescents. Various studies have found that the lifetime prevalence of NSSI in nonclinical adolescents is as high as 17–23% [4,5], and NSSI among nonclinical adolescents constitutes a major public health concern.

Given that even a single incident of NSSI remains a harmful dysregulated behavior that has a long-lasting and detrimental effect on an adolescent’s well-being [6], numerous studies have been dedicated to finding the risk factors of NSSI initiation among adolescents [7,8,9,10]. Moreover, the majority of adolescents who conduct NSSI finally remit the behavior [11]. For instance, among adolescents who reported NSSI between 13 and 17 years of age, nearly half of them reported NSSI cessation within a year [8,12]. Thus, examination of the predictors of NSSI cessation also constitutes an area of great interest [8,9,13]. An understanding of the psychosocial correlates of NSSI initiation and cessation not only helps to identify adolescents at risk for new engagement in NSSI and to design early preventative interventions for these groups, but also to elucidate key factors that could be targeted for therapy.

According to the integrated theoretical model [1], the factors increasing the risk of NSSI fall into two broad domains: Interpersonal and intrapersonal. On the one hand, adolescents could be motivated or triggered to engage in NSSI by their interpersonal risk factors [14]. On the other hand, intrapersonal risk factors may also predispose adolescents to respond to stress with affective dysregulation, creating a need to use NSSI [15]. Indeed, previous studies conducted among nonclinical adolescents have provided empirical support for each of these two domains in the initiation of NSSI engagement.

Specifically, the chances of self-injurious behavior initiation increase with interpersonal risks, including a negative relationship with parents [16], peer victimization [10], low social support [7], and stressful life events [17]. Moreover, intrapersonal risks such as low self-esteem [9], negative affect [18], depression and anxiety [8], and problem behavior [19] have also been suggested as predictors of NSSI initiation among adolescents. Using a longitudinal design, these studies have provided insight into the factors possibly underlying NSSI initiation. However, it should be noted that these longitudinal studies offer conflicting results about which risk factors are associated with NSSI initiation. For instance, Cassels et al. [19] found that behavioral problems predicted NSSI increase over one year, whereas Koenig et al. [8] did not find a significant longitudinal association between behavioral problems and NSSI initiation.

With regard to NSSI cessation, empirical evidence has suggested that many factors associated with NSSI initiation also precede NSSI cessation [20], but there may exist unique factors, intrapersonal factors in particular, in predicting the cessation of NSSI. For example, researchers have found that depression and anxiety are significantly associated with subsequent cessation of NSSI (cease NSSI vs. maintain NSSI), but are not associated with the initiation of NSSI (started to engage in NSSI vs. never engaged in NSSI) among adolescents [8]. Other researchers have also found that lack of perseverance is a unique predictor of NSSI cessation [21]. While these studies imply that potential differences exist between the mechanisms of NSSI initiation and NSSI cessation, few studies have examined the factors related to initiation and cessation simultaneously. Studies of NSSI cessation mainly compare individuals who have ceased their NSSI behavior to those who have maintained NSSI behavior, revealing that negative emotions [22] and resilience [13] could differentiate between current and past self-injurers. A few studies using a longitudinal design among adolescents have found that greater impulsive behavior [12], higher levels of self-esteem, and family support [9] are associated with subsequent cessation of NSSI relative to maintenance.

Overall, many studies have examined the factors associated with NSSI initiation and cessation. However, most of these studies included only a narrow set of factors and lacked systematic exploration integrating both initiation and cessation of NSSI, which may limit the identification of key factors in predicting different processes of NSSI. Additionally, previous studies have been mainly conducted among Western samples, and findings from these studies may not be generalizable to Eastern cultures. Specifically, the influence of Chinese culture on this relationship is mainly two-pronged. On the one hand, since a collectivistic culture is predominant in China, interpersonal connectedness is highly valued [23] and the interpersonal model [24] may be a particularly relevant framework for understanding NSSI development among Chinese adolescents. Previous studies conducted among Chinese adolescents have found that interpersonal factors such as peer victimization [25], negative life events, and less social support [26] are relevant risk factors for NSSI. On the other hand, the Chinese Confucian culture emphasizes individuals’ positive attitude when facing adversity [27]. Given that stress is an important factor regarding NSSI, such beliefs about adversity held by Chinese adolescents may be more salient when predicting NSSI. Taken together, further research is warranted to examine longitudinal associations between a comprehensive range of factors and NSSI initiation and cessation among Chinese adolescents.

Using a two-wave longitudinal design, the current study aimed to examine the association between a broad range of interpersonal and intrapersonal potential predictors and subsequent NSSI initiation and cessation over six months among nonclinical Chinese adolescents. Based on previous evidence, it was hypothesized that both interpersonal and intrapersonal risks would be associated with NSSI initiation, while intrapersonal factors would be relevant in NSSI cessation. Specifically, we expected that adolescents with higher levels of peer victimization, stressful life events, loneliness, depression, problem behavior, and lower levels of social support and self-esteem would be more likely to initiate NSSI behavior six months later. We also hypothesized that intrapersonal factors such as self-esteem, loneliness, and depression would be associated with adolescents’ NSSI cessation. Given the individual positive attitude when facing adversity emphasized in Chinese culture, we expected that adolescents’ belief about adversity would be associated with both the initiation and cessation of NSSI.

## 2. Methods

### 2.1. Participants

Participants were recruited from the 7th and 8th grades in public junior high schools in Guizhou, China to participate in a 6-month longitudinal study. In sampling the participants, we randomly contacted public junior high schools in Guizhou to ask their willingness to participate in this study. Four junior high schools agreed to participate, and their school principals gave permission. The overall sample consisted of 938 adolescents (496 females, 52.9%; age range 12–16 years, mean age = 13.48, SD = 0.98) who completed self-report questionnaires at both baseline and at the 6-month follow-up. The participants reported the highest level of education completed by their parents. A total of 28.1% of the parents had completed primary school, 61.4% had completed secondary school, and 10.4% had an education level of high school or above.

### 2.2. Procedure

Data were collected in December 2017 (baseline) and June 2018 (follow-up), following the same protocol. First, written informed consent was sent to the adolescents and their primary caregivers before the research. Only those who agreed to participate in the research were included. Then, during regular school hours, the students completed the questionnaires in the classroom. Each class was accompanied by at least one trained researcher who was available to answer any questions related to the survey during the entire session. The completion of the questionnaire lasted approximately 40 min. After the survey, each participant received a gift (a pen and a notebook at baseline and a t-shirt at follow-up) for their participation. This study was approved by the Research Commission of Beijing Normal University and the principals of the participating schools.

### 2.3. Measures

All participants completed self-reported questionnaires to measure their NSSI at baseline and follow-up. Additionally, a wide range of interpersonal and intrapersonal factors and demographic characteristics were assessed at baseline. The selection of interpersonal and intrapersonal factors was informed by the integrated theoretical model of NSSI [1] and by previous empirical literature. Interpersonal factors tapped childhood/adolescence adversities, positive relationships, parenting, and stressful life events, while the intrapersonal factors tapped temperaments, personalities, and externalizing and internalizing problems. All study variables were assessed by the translated Chinese version of scales that had been successfully used in samples of Chinese children and adolescents with reliability and validity.

NSSI. Adolescents’ non-suicidal self-injury was assessed using a shortened and modified version [28] of the Deliberate Self-Harm Inventory (DSHI) [29]. Adolescents were asked to report the frequency of nine self-injury items (e.g., knocking, scratching, piercing, burning, biting, and cutting to bleed) on a scale of 1 (never) to 5 (always) during the past six months. In our sample, the Cronbach’s α was 0.80 and 0.86 at baseline and follow-up, respectively. Participants’ responses were then recoded to a binary categorical variable. That is, participants who reported 1 (never) for all nine self-injury items were coded as 0 (no NSSI); participants who reported 2 (seldom) to 5 (always) for any self-injury item were coded as 1 (have NSSI).

Childhood/adolescence adversities. The parent-to-child version of the Conflict Tactics Scale (CTSPC) [30] consists of 18 items to assess childhood maltreatment such as psychological aggression, corporal punishment, physical abuse, and severe physical abuse during the past year. The Cronbach’s α was 0.84. The frequency was used as an index of maltreatment (the full range was 0–25). The Chinese-adapted version [31] of the Multidimensional Peer-Victimization Scale (MPVS) [32] consists of 21 items to assess peer victimization, scored on a scale of 0 (never) to 3 (always). The Cronbach’s α was 0.92.

Social support. Social support, including family support, teacher support, and friend support, was measured with the Chinese-adapted version [33] of the Multidimensional Scale of Perceived Social Support (MSPSS) [34], which consists of 12 items scored on a scale of 1 to 5. The Cronbach’s α was 0.83. Friendship quality was measured with a subscale of the Chinese version [35] of the Network of Relationships Inventory [36], which consists of 3 items scored on a scale of 1 to 5. The Cronbach’s α was 0.58.

Parenting. Parental cohesion was measured with the Family Adaptability and Cohesion Scale II (FACS-II) [37], which consists of 6 items each for fathers and mothers scored on a scale of 1 (never) to 5 (always). The Cronbach’s α was 0.87. The 18-item Parental Psychological Control Scale [38] and the 6-item modified version of the Parental Behavioral Control Scale [39] were used to assess psychological control and behavioral control for mothers and fathers, respectively. Psychological control tapped guilt induction, love withdrawal, and authority assertion on a 5-point scale ranging from 1 (strongly disagree) to 5 (strongly agree). The Cronbach’s α was 0.87 for both the father and mother subscales. Behavioral control tapped parental knowledge of youth behavior on a 3-point scale ranging from 1 (doesn’t know) to 3 (knows a lot). The Cronbach’s α was 0.72 and 0.80 for the father and mother subscales, respectively.

Recent stressful life events. The Adolescent Self-Rating Life Events Checklist [40] designed for Chinese adolescents was used to assess the stressful life events experienced by adolescents during the last year. The checklist included interpersonal (5 items), academic (5 items), punishment (6 items), loss (3 items), adaptation (4 items), and other events (3 items) with a potential for causing distress among adolescents. The items were scored on a scale of 1 (not at all) to 5 (very seriously), or were coded as 0 to indicate that the event had not happened to them. The Cronbach’s α was 0.89.

Personalities. The Chinese Beliefs about Adversity Scale [27] and the Chinese version of the Self-Esteem Scale [41] were used to assess adolescents’ beliefs about adversity (9 items) and self-esteem (10 items). The Cronbach’s α was 0.70 and 0.68, respectively. For beliefs about adversity, items were scored on a scale of 1 (strongly disagree) to 6 (strongly agree). For self-esteem, items were scored on a scale of 1 (strongly disagree) to 5 (strongly agree).

Behavior. Sixteen items drawn from the Chinese-adapted version [42] of the Youth Self-Report (YSR) [43] were used to assess adolescents’ externalizing problems, including problem behavior (12 items) and prosocial behavior (4 items). The items were scored on a scale of 1 (never) to 4 (very often). The Cronbach’s α was 0.81.

Emotions. The Student’s Life Satisfaction Scale [44], the Chinese Student’s Positive And Negative Affect Scale [45], the pure loneliness subscale of the Chinese version [46] of the Child Loneliness Scale [47], and the Center for Epidemiologic Studies Depression Scale for Children (CES-DC) [48] were used to assess adolescents’ internalizing problems, including life satisfaction (7 items), positive affect (8 items), negative affect (6 items), loneliness (6 items), and depression (20 items). The Cronbach’s α was 0.68, 0.80, 0.79, 0.88, and 0.84, respectively. For life satisfaction, the items were scored on a scale of 1 (strongly disagree) to 5 (strongly agree). For positive affect, negative affect, and loneliness, the items were scored on a scale of 1 (never/strongly disagree) to 4 (always/strongly agree). For depression, the items were scored on a scale of 0 (never) to 3 (always).

Demographic variables. The demographic information obtained from the participants included gender, age, grade, and left-behind or non-left-behind. We adopted the MacArthur Scale to assess participants’ subjective socioeconomic status [49]. Participants were presented with a diagram of a “social ladder” with 10 rungs from 1 to 10 and were asked to rank themselves by choosing the rung that represented their relative position in society. Higher scores indicate higher subjective socioeconomic status.

### 2.4. Data Analysis

Data were analyzed using SPSS 21 and Mplus 7.11. The proportions of missing values ranged from 0.2% to 10.9% for each study variable. For scales with at least 60% completed items, missing items were replaced with the mean score of the answered items [50], reducing the proportions to 0.1–4.9%. Then, the proportions of the missing variables for each participant were calculated. If 10% or more of the predictor variables were missing or the outcome variables were missing, the participant was excluded from the analysis. Finally, a sample of 913 adolescents were entered into the subsequent analysis, with 625 reporting no NSSI at baseline and 288 reporting NSSI at baseline. Based on Markov Chain Monte Carlo method [51], multiple imputation was used to generate 10 imputed datasets for each exposure of interest within Mplus.

In all analyses, the dependent variable was dichotomous: With NSSI or without NSSI. Descriptive analyses were conducted to describe the demographic (i.e., gender, age, left-behind, and subjective socioeconomic status (SES)) and descriptive (percentage, mean, and SD) characteristics of the study samples. Binary logistic regression was used to determine how well the predictors distinguished between adolescents with NSSI and those without NSSI at the 6-month follow-up. Analyses were conducted among the participants who reported no NSSI (*n* = 625) at baseline to predict NSSI initiation, and among participants who reported NSSI (*n* = 288) at baseline to predict NSSI cessation. First, bivariate analyses were performed to examine the associations between each potential predictor and NSSI at follow-up. For bivariate analyses, the false discovery rate, suggested by Benjamini and Hochberg [52], was used to correct for multiple testing. Next, significant predictors in the bivariate analyses were entered into a multi-predictor logistic regression because our aim was to determine the independent contribution of each predictor on NSSI when controlling for the impact of other potential predictors. Continuous predictors were standardized before being entered into the regression. Odds ratios (OR) and their 95% confidence intervals (CIs) were estimated. All tests were two-tailed, and statistical significance was evaluated with an α level of significance of 0.05.

## 3. Results

The demographic characteristics of the samples are shown in Table 1. Among the subsample with no NSSI at baseline (*n* = 625), 24.3% (*n* = 152) reported conducting NSSI at the six-month follow-up. No significant differences in the demographic characteristics except for subjective SES were found between the future NSSI and no NSSI subgroups. Participants who reported a transition to NSSI had higher levels of subjective SES than those that remained without NSSI. Among the subsample with NSSI at baseline (*n* = 288), 33.3% (*n* = 96) reported recovering from NSSI at the six-month follow-up. No significant differences in demographic characteristics except for gender were found between the future NSSI and no NSSI subgroups. Participants who reported cessation of NSSI were more likely to be male. With the aim of excluding potentially confounding effects, subjective SES was controlled in the prediction of NSSI initiation and gender was controlled in the prediction of NSSI cessation.

Table 2 shows the associations between each predicting factor and future NSSI among the subsample with no NSSI at baseline. In the bivariate logistic regression models, all factors but childhood adversities, friend support, teacher support, and life satisfaction were significantly associated with NSSI initiation. Specifically, higher levels of family support, parental cohesion, parental behavioral control, positive beliefs about adversity, self-esteem, prosocial behavior, and positive affect were associated with reduced odds of reporting NSSI at follow-up. The odds ratios ranged from 0.96 (prosocial behavior) to 0.93 (beliefs about adversity and self-esteem). Higher levels of peer victimization, parental psychological control, all six kinds of recent stressful life events, problem behavior, negative affect, loneliness, and depression were associated with increased odds of reporting NSSI at follow-up. The odds ratios ranged from 1.04 (maternal psychological control and adapt stressful life events) to 1.10 (problem behavior and loneliness). The results were unchanged when adjusting for multiple testing.

Then, all significant predictors were entered into a multi-predictor logistic regression, as shown in Table 3. The significant effects of all of the interpersonal factors observed in the bivariate model were no longer significant, whereas intrapersonal factors such as beliefs about adversity (OR = 0.96, 95% CI: 0.92–0.99), problem behavior (OR = 1.10, 95% CI: 1.04–1.15), prosocial behavior (OR = 1.05, 95% CI: 1.00–1.10), and loneliness (OR = 1.06, 95% CI: 1.01–1.11) remained significant in predicting NSSI initiation.

Table 4 shows the associations between each predicting factor and future recovery from NSSI among the subsample with NSSI at baseline. In the bivariate logistic regression models, only a few factors were significantly associated with NSSI cessation. Specifically, a higher frequency of corporal punishment (OR = 0.95, 95% CI: 0.90–1.00), higher levels of peer victimization (OR = 0.95, 95% CI: 0.90–1.00), three kinds of recent stressful life events (ORs = 0.92–0.94), problem behavior (OR = 0.94, 95% CI: 0.88–0.99), and life satisfaction (OR = 0.93, 95% CI: 0.88–0.98) were associated with reduced odds of recovering from NSSI. However, after adjusting for multiple comparisons, these associations were no longer significant. Therefore, multi-predictor logistic regression was not performed for NSSI cessation.

## 4. Discussion

This study aimed to identify which interpersonal and intrapersonal factors prospectively predicted the initiation and cessation of NSSI in a community sample of Chinese adolescents over six months. First, the preliminary results found that 24.3% of the adolescents who reported no NSSI at baseline started to engage in NSSI at follow-up, and 33.3% of the adolescents who reported NSSI engagement at baseline ceased NSSI at follow-up. Although it was not the main goal of our study, the results supported the view that NSSI among adolescents may be transient, which provides a foundation for subsequent research. Second, we identified interpersonal and intrapersonal factors that predicted future NSSI behavior in different subgroups. Among participants without NSSI at baseline, future NSSI engagement was associated with both interpersonal and intrapersonal factors in the bivariate analyses, while only the intrapersonal factors (i.e., beliefs about adversity, loneliness, problem behavior, and prosocial behavior) remained significant in the multi-predictor model. However, among the participants with NSSI at baseline, inconsistent with our hypothesis, few predicting factors were associated with future NSSI cessation. Finally, comparing the predictors found in previous research with Western samples to those found in our study, we found differences that may guide the development of early prevention, intervention, and therapy for NSSI among Chinese adolescents.

As we hypothesized, the results of the bivariate analyses indicated that both interpersonal and intrapersonal factors are relevant in NSSI development among Chinese adolescents. Concerning interpersonal factors, the significant predictors, including peer victimization, stressful life events (e.g., misunderstood by others and conflict with classmates), lack of family support, and lack of parental cohesion, pertain to negative interpersonal experiences. These findings are consistent with prior research among Western samples, which found that interpersonal stressors prospectively predict new engagement in NSSI behavior [10,17,53]. When faced with negative events, adolescents may experience emotional distress [54], which is an undesired experience that adolescents would try to avoid. Indeed, NSSI is one effective way of regulating emotions and reducing tension [55,56]. Thus, adolescents are likely to conduct self-injurious behavior to cope with emotional distress and to reduce tensions.

However, the interpersonal factors were attenuated (toward null) after the intrapersonal factors were included in the multi-predictor model; only loneliness, beliefs about adversity, and problem and prosocial behavior were strongly associated with NSSI initiation. Previous studies have found a similar trend when controlling for intrapersonal factors [57,58], suggesting that intrapersonal factors such as negative emotions are the proximal risks for NSSI behavior. However, the relevant intrapersonal factors in our study may differ from those found in Western samples. First, relative to depression, identified as a risk factor for NSSI in Western samples [8,59], loneliness, instead of depression, was the relevant factor found in the current study. On the one hand, loneliness represents a negative emotional outcome of unsatisfied interpersonal needs and, as such, supports the interpersonal model [24] in the Chinese collectivistic culture. On the other hand, the results indicated that only those interpersonal events with actual negative outcomes increase the probability of self-injury.

Second, in contrast to the significant role of self-esteem among Western samples [7,9], our study indicated that beliefs in adversity are the more important intrapersonal factor in reducing the odds of engaging in NSSI behavior among Chinese adolescents. Previous research conducted among Chinese adolescents found that higher levels of beliefs in adversity are directly associated with less negative emotions longitudinally [60]. Considering that negative emotions may further increase one’s engagement in NSSI [55], it is possible that adolescents who endorse positive beliefs about adversity may have cognitive resources that help them to think positively and thus to reduce negative emotions, which leads to a lower tendency to initiate NSSI behavior. Future research is encouraged to explore the underlying mechanisms between interpersonal/intrapersonal factors and NSSI behavior across different cultures.

In addition, in line with the emerging literature [16,61] about the association between externalizing behavior and self-injurious behavior, our study demonstrated that problem behavior was a robust predictor of NSSI initiation, and this prediction further expanded over the longitudinal time period. Specifically, more problem behavior and less prosocial behavior were associated with higher odds of starting to engage in NSSI relative to never having engaged in NSSI. According to the life history theory, for adolescents who are exposed to an unsafe and unpredictable environment, maintaining basic functioning, instead of growth, is their priority [62]. When calibrating, rather than developing, to match the harsh environment, adolescents may develop risky behavior such as crime and aggression [63]. Recent research shows that self-harming behavior is also a typical type of such risky behavior [64]. Considering that both problem behavior and self-injurious behavior represent adolescents’ functional adaptation to stressful environments, it is possible that adolescents who engage in problem behavior are more likely to engage in NSSI behavior.

In predicting NSSI cessation, although a few bivariate associations were found, none of them remained significant after multiple testing. Although unexpected, these findings support the view that NSSI is a complex behavior and is multidetermined. While it can be triggered by various kinds of negative environments or initiated by intrapersonal vulnerabilities [1,14], the reason for ceasing NSSI behavior seems to be more complicated. Compared to interpersonal and intrapersonal factors, the characteristics of NSSI behavior may be the more relevant factor in predicting NSSI cessation. For example, previous studies found that the less individuals conducted NSSI behavior, the more likely it was for them to cease NSSI [57], even after controlling for intrapersonal as well as interpersonal factors [58].

The current findings should be considered in light of several limitations. We included adolescents who had and who had not engaged in NSSI at baseline as two separate samples so that we could examine the predictors of change of NSSI during the study period. Although we consider this approach to be a strength of the study, it may have weakened the associations with some of our predictors. For example, childhood adversities are a widely recognized risk factor associated with NSSI [1], so adolescents with childhood adversities would likely have already engaged in NSSI at baseline, and therefore showed no significant variation in NSSI behavior at follow-up. Moreover, our study regarded NSSI as detrimental behavior that could be triggered by risky interpersonal factors, predisposed by intrapersonal vulnerabilities. However, NSSI may also be a typical type of the fast life history strategy, characterized by seeking immediate gratification over long-term goals [64], which represents adolescents’ effort to adapt to an environment that is unsafe and unpredictable [63]. Future research needs to examine the adaptive functions of NSSI behavior and to try to find strategies that yield outcomes comparable to those achieved by self-harming behavior and also with long-term benefits.

## 5. Conclusions

Despite the limitations of this study, the findings support that both interpersonal and intrapersonal factors are prospectively associated with NSSI initiation, and emphasize the particular relevance of intrapersonal factors (e.g., loneliness and beliefs about adversity) among a Chinese adolescent sample. There are important potential implications for the intervention and treatment of NSSI. The findings herein highlight the significance of implementing programs to teach adolescents adaptive coping strategies to deal with interpersonal stressful situations and to reduce their sense of loneliness, thereby reducing the risk of engaging in NSSI behavior. Cultivating positive attitudes toward negative events may also be protective against NSSI among Chinese samples. Additionally, externalizing behavior (i.e., problem and prosocial behavior) may be a relevant antecedent of NSSI, which calls for further research regarding the underlying mechanism. Moreover, it may be valuable to consider differences in the risk process for the initiation of NSSI and its maintenance. Interventions targeting initiation should focus on intrapersonal factors under the context of a specific culture, while more research is needed to identify relevant factors in promoting the cessation of NSSI.

## Figures and Tables

**Table 1 ijerph-17-09454-t001:** Demographic characteristics for the study samples.

Demographic Characteristics	Total Sample	No NSSI at Baseline *N* = 625		χ2/t	NSSI at Baseline *N* = 288		χ2/t
		T2 No (*n* = 473)	T2 Yes (*n* = 152)		T2 No (*n* = 96)	T2 Yes (*n* = 192)	
Gender							
Male	431 (47.2%)	228 (48.2%)	71 (46.7%)	0.10	54 (56.3%)	78 (40.6%)	6.29 *
Female	482 (52.8%)	245 (51.8%)	81 (53.3%)		42 (43.7%)	114 (59.4%)	
Age	13.48 ± 0.97	13.53 ± 0.96	13.40 ± 1.01	1.45	13.52 ± 1.01	13.42 ± 0.96	0.81
Left-behind							
No	288 (31.5%)	150 (31.7%)	46 (30.3%)	0.41	31	61	0.02
Yes	581 (63.6%)	295 (62.4%)	103 (67.8%)		60	123	
Subjective SES	4.36 ± 1.26	4.28 ± 1.22	4.55 ± 1.38	−2.28 *	4.37 ± 1.21	4.38 ± 1.28	−0.09

Note: Data are *N* (percentages) or mean ± SD. * *p* < 0.05. NSSI, non-suicidal self-injury; SES, socioeconomic status.

**Table 2 ijerph-17-09454-t002:** Bivariate associations between the predictors and NSSI initiation.

	No Transition to NSSI	Transition to NSSI	OR ^1^	95% CI	*p*
**Interpersonal factors**					
Childhood/adolescence adversities					
Psychological aggression	1.56 (2.90)	1.89 (3.12)	1.02	0.99–1.06	0.168
Corporal punishment	0.48 (1.54)	0.53 (1.02)	1.01	0.98–1.05	0.514
Physical abuse	0.41 (2.23)	0.34 (1.09)	1.00	0.96–1.03	0.839
Severe physical abuse	0.07 (0.55)	0.08 (0.39)	1.01	0.97–1.04	0.703
Peer victimization	0.47 (0.40)	0.62 (0.39)	1.07	1.04–1.11	0.000 *
Positive relationships					
Family support	4.19 (0.68)	4.03 (0.78)	0.95	0.92–0.99	0.014 *
Friend support	3.96 (0.81)	3.86 (0.89)	0.97	0.94–1.01	0.128
Teacher support	3.97 (0.87)	3.88 (0.97)	0.98	0.94–1.01	0.163
Parenting					
Paternal cohesion	4.03 (0.78)	3.80 (0.87)	0.95	0.92–0.98	0.001 *
Maternal cohesion	4.20 (0.69)	3.94 (0.82)	0.94	0.90–0.97	0.000 *
Paternal behavioral control	2.22 (0.46)	2.09 (0.49)	0.95	0.92–0.98	0.004 *
Maternal behavioral control	2.30 (0.48)	2.15 (0.54)	0.95	0.91–0.98	0.001 *
rePaternal psychological control	2.66 (0.68)	2.84 (0.69)	1.05	1.01–1.08	0.023 *
Maternal psychological control	2.70 (0.69)	2.86 (0.71)	1.04	1.01–1.08	0.013 *
Recent stressful life events					
Interpersonal	1.15 (0.81)	1.49 (0.82)	1.08	1.05–1.12	0.000 *
Academic	1.58 (0.85)	1.88 (0.85)	1.08	1.04–1.11	0.000 *
Punishment	0.74 (0.64)	0.99 (0.67)	1.08	1.04–1.11	0.000 *
Loss	0.77 (0.95)	1.06 (0.96)	1.06	1.02–1.10	0.001 *
Adaptation	0.67 (0.62)	0.81 (0.57)	1.04	1.01–1.08	0.010 *
Other	0.33 (0.59)	0.49 (0.73)	1.05	1.01–1.08	0.006 *
**Intrapersonal factors**					
Personality					
Beliefs about adversity	5.08 (0.57)	4.82 (0.72)	0.93	0.90–0.96	0.000 *
Self-esteem	3.41 (0.58)	3.19 (0.56)	0.93	0.90–0.96	0.000 *
Behavior					
Problem behavior	1.40 (0.27)	1.55 (0.33)	1.10	1.06–1.14	0.000 *
Prosocial behavior	2.99 (0.64)	2.85 (0.71)	0.96	0.93–0.99	0.017 *
Emotions					
Life satisfaction	3.10 (0.71)	3.11 (0.80)	1.00	0.96–1.03	0.765
Positive affect	3.03 (0.56)	2.81 (0.65)	0.93	0.90–0.96	0.008 *
Negative affect	1.91 (0.61)	2.14 (0.67)	1.07	1.04–1.11	0.000 *
Loneliness	1.60 (0.65)	1.96 (0.78)	1.10	1.07–1.14	0.000 *
Depression	16.38 (8.28)	20.30 (9.00)	1.09	1.05–1.13	0.000 *

^1^ The odds ratios (ORs) were estimated based on a separate model for each predictor, adjusted for socioeconomic status. * Significant after false discovery rate correction.

**Table 3 ijerph-17-09454-t003:** Associations between predictors and NSSI initiation in the multi-predictor model.

	OR ^1^	95% CI	*p*
**Interpersonal factors**			
Peer victimization	0.99	0.95–1.03	0.581
Family support	1.01	0.97–1.05	0.820
Paternal cohesion	1.01	0.96–1.06	0.716
Maternal cohesion	0.99	0.94–1.04	0.591
Paternal behavioral control	1.00	0.95–1.06	0.975
Maternal behavioral control	0.99	0.94–1.05	0.786
Paternal psychological control	1.07	0.96–1.20	0.220
Maternal psychological control	0.94	0.84–1.04	0.238
Recent stressful life events			
Interpersonal	1.01	0.96–1.06	0.709
Academic	1.03	0.98–1.08	0.292
Punishment	1.02	0.97–1.07	0.484
Loss	1.04	1.00–1.08	0.054
Adaptation	0.98	0.94–1.02	0.268
Other	0.98	0.94–1.02	0.296
**Intrapersonal factors**			
Beliefs about adversity	0.96	0.92–0.99	0.027
Self-esteem	0.99	0.95–1.03	0.667
Problem behavior	1.10	1.04–1.15	0.000
Prosocial behavior	1.05	1.00–1.10	0.034
Positive affect	0.98	0.94–1.03	0.404
Negative affect	0.99	0.94–1.03	0.566
Loneliness	1.06	1.01–1.11	0.019
Depression	1.01	0.96–1.06	0.835

^1^ The ORs were estimated in a model including all significant predictors in the previous bivariate analyses, adjusted for socioeconomic status.

**Table 4 ijerph-17-09454-t004:** Bivariate associations between the predictors and NSSI cessation.

	Maintained NSSI	Ceased NSSI	OR ^1^	95% CI	*p*
**Interpersonal factors**					
Childhood/adolescence adversities					
Psychological aggression	3.22 (4.20)	2.91 (4.12)	0.99	0.94–1.05	0.752
Corporal punishment	1.27 (2.26)	0.72 (1.49)	0.95	0.90–1.00	0.040
Physical abuse	1.03 (2.69)	0.66 (1.76)	0.96	0.90–1.01	0.103
Severe physical abuse	0.25 (0.93)	0.22 (0.88)	0.99	0.93–1.04	0.611
Peer victimization	0.76 (0.46)	0.65 (0.41)	0.95	0.90–1.00	0.046
Positive relationships					
Family support	3.80 (0.84)	3.85 (0.84)	1.01	0.95–1.06	0.841
Friend support	3.67 (0.84)	3.81 (0.77)	1.04	0.98–1.10	0.170
Teacher support	3.58 (0.93)	3.72 (0.95)	1.02	0.97–1.08	0.419
Parenting					
Paternal cohesion	3.61 (0.90)	3.76 (0.82)	1.03	0.98–1.09	0.259
Maternal cohesion	3.74 (0.87)	3.88 (0.76)	1.03	0.98–1.09	0.224
Paternal behavioral control	2.00 (0.47)	2.07 (0.41)	1.03	0.98–1.09	0.213
Maternal behavioral control	2.06 (0.52)	2.13 (0.47)	1.03	0.98–1.09	0.224
Paternal psychological control	2.85 (0.72)	2.93 (0.68)	1.02	0.96–1.07	0.556
Maternal psychological control	2.92 (0.70)	3.00 (0.69)	1.02	0.96–1.08	0.486
Recent stressful life events					
Interpersonal	1.83 (0.85)	1.53 (0.75)	0.93	0.88–0.98	0.007
Academic	2.01 (0.81)	1.84 (0.88)	0.97	0.91–1.02	0.213
Punishment	1.28 (0.77)	1.15 (0.77)	0.96	0.91–1.01	0.151
Loss	1.25 (1.09)	1.03 (0.90)	0.95	0.90–1.00	0.057
Adaptation	1.04 (0.72)	0.83 (0.75)	0.94	0.89–0.99	0.017
Other	0.85 (0.92)	0.69 (0.78)	0.93	0.88–0.99	0.014
**Intrapersonal factors**					
Personality					
Beliefs about adversity	4.75 (0.65)	4.70 (0.70)	0.99	0.94–1.05	0.712
Self-esteem	3.00 (0.58)	3.13 (0.53)	1.01	0.99–1.10	0.137
Behavior					
Problem behavior	1.66 (0.37)	1.61 (0.31)	0.94	0.88–0.99	0.024
Prosocial behavior	2.84 (0.62)	2.83 (0.60)	1.01	0.96–1.07	0.623
Emotions					
Life satisfaction	3.13 (0.71)	2.90 (0.78)	0.93	0.88–0.98	0.009
Positive affect	2.76 (0.55)	2.80 (0.60)	1.01	0.96–1.07	0.636
Negative affect	2.39 (0.65)	2.33 (0.59)	0.98	0.93–1.04	0.566
Loneliness	2.06 (0.79)	1.97 (0.73)	0.98	0.93–1.04	0.463
Depression	24.66 (9.60)	22.54 (8.94)	0.96	0.91–1.01	0.147

^1^ The ORs were estimated based on a separate model for each predictor, adjusted for gender.

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
