# Peer review of "Prospective Interpersonal and Intrapersonal Predictors of Initiation and Cessation of Non-Suicidal Self-Injury among Chinese Adolescents"

_ijerph, 2020, doi:10.3390/ijerph17249454_

Round 1

Reviewer 1 Report

The submitted manuscript intends to shed light into the prevention of self-injury focusing in a specific population: adolescents in China. Authors made a big effort to systematically observe the impact of different interpersonal and intrapersonal variables that might have a relationship with self-injury behaviours. They include a considerable sample (nearly a thousand adolescents) that provided detailed information about all the relevant variables via self-report. However, no other informants different from the adolescents or any other objective complementary measure is acquired, being one of the main weakness that limit the extent to which the results might be interpreted. All in all, the present paper provides a valuable research, although some aspects might be addressed:

  1. Authors stated that Chinese society had some characteristics that may modulate the association between explored predictors and self-injury. I suggest authors to review literature that may support that claim.
  2. I encourage authors to formulate hypotheses about expected results regarding the reviewed literature.
  3. For the bivariate regressions, multiple comparisons correction must be performed.
  4. Given the sample size, I should recommend authors to attempt to explore a statistical model on how resulting relevant variables relate to predict self-injury and try to determine the contribution of each variable in the model.
  5. In would suggest authors to compare the difference in the predictors found in previous research with Western samples to that found in their study.

Author Response

1. Authors stated that Chinese society had some characteristics that may modulate the association between explored predictors and self-injury. I suggest authors to review literature that may support that claim.

Response: Thank you very much for your suggestion. We have thoroughly reviewed the literature and added the relevant support for this claim. Please see the revision in the Introduction section, line 91-100.

2. I encourage authors to formulate hypotheses about expected results regarding the reviewed literature.

Response: Thank you very much. Following your advice, we have added hypotheses about expected results regarding the reviewed literature in Introduction section, line 105-119.

3. For the bivariate regressions, multiple comparisons correction must be performed.

Response: Following your advice, we have performed multiple comparisons correction for the bivariate regressions. The results were revised accordingly by adding note in Table 2 and in line 96 and line 314-317.

4. Given the sample size, I should recommend authors to attempt to explore a statistical model on how resulting relevant variables relate to predict self-injury and try to determine the contribution of each variable in the model.

Response: Thank you very much for your valuable suggestion. Following your advice, we have explored the multivariate model including all the variables that were significant in the bivariate analyses.

5. I would suggest authors to compare the difference in the predictors found in previous research with Western samples to that found in their study.

Response: Thank you for your suggestion. We have provided a fully discussion in the discussion part to compare the difference in the predictors found in previous research with Western samples to that found in our study.

Reviewer 2 Report

The manuscript “Prospective Interpersonal and Intrapersonal 2 Predictors of Initiation and Cessation of Non-Suicidal Self-Injury among Chinese Adolescents” describes a longitudinal , two-wave survey study of 913 adolescents on NSSI initiation and cessation. The analytic method is very briefly described and is unclear- seems to rely on logistic regressions for individual risk factors, without trying to test an overall model for NSSI, the results are mostly in the form of long tables with no multipredictor models applied.. Additionally, the outcome is measures on a 5-point scale and yet the analytic method describes logistic regression models for NSSI initiation or cessation without specifying how these were defined. A journal-provided paragraph was also left in the Methods section. Overall, while the data is very interesting, this manuscript needs more careful work to present and analyze the data appropriately, before resubmission.

Other Comments:

  1. The Methods section start with a no-doubt journal provided standard paragraph about what should be included in a Methods section, that was left in the manuscript. Please remove.
  2. The multiple imputation seems appropriate and the missingness reasonable, but please provide the exact imputation method used.
  3. Line 194 statistical terminology unclear: Bivariate logistic regression=binary logistic regression, or logistic regression with a single predictor?
  4. The outcome variable (NSSI at 6 month) is quantitative-how was then “NSSI initiation or cessation defined? How was the frequency of it used- was a proportional odds logistic regression model used? If so, this should be described in the Methods section and again mentioned in the results sectioon
  5. Were the models single predictor, adjusted only for gender at SE? What does the “adjusted analyses” referred to in line 211 mean?

Language mistakes are on mild to moderate level- stopped correcting them after line 130.

  1. Grammar mistake: Line 100, “Four junior high schools agree” should be “Four junior high schools agreed”
  2. Extra comma on line 118: “Besides, a wide range of interpersonal and intrapersonal factors”
  3. Line 123: “studying variables” should be “study variables, “and had been successfully used” should be “that had been successfully used”

Author Response

The manuscript “Prospective Interpersonal and Intrapersonal 2 Predictors of Initiation and Cessation of Non-Suicidal Self-Injury among Chinese Adolescents” describes a longitudinal, two-wave survey study of 913 adolescents on NSSI initiation and cessation. The analytic method is very briefly described and is unclear- seems to rely on logistic regressions for individual risk factors, without trying to test an overall model for NSSI, the results are mostly in the form of long tables with no multipredictor models applied. Additionally, the outcome is measures on a 5-point scale and yet the analytic method describes logistic regression models for NSSI initiation or cessation without specifying how these were defined. A journal-provided paragraph was also left in the Methods section. Overall, while the data is very interesting, this manuscript needs more careful work to present and analyze the data appropriately, before resubmission.

Response: Thank you very much for your valuable suggestions. We have carefully revised the manuscript by adding more information about the analytic method, applying multi-predictor models in predicting NSSI, and presenting the data appropriately.

Other Comments:

1.The Methods section start with a no-doubt journal provided standard paragraph about what should be included in a Methods section, that was left in the manuscript. Please remove.

Response: Thank you very much for correction. we have deleted the paragraph in the Methods section.

2. The multiple imputation seems appropriate and the missingness reasonable, but please provide the exact imputation method used.

Response: Thank you. We have provided the exact imputation method used in 2.4 Data Analysis.

3. Line 194 statistical terminology unclear: Bivariate logistic regression=binary logistic regression, or logistic regression with a single predictor?

Response: Thank you. “Bivariate logistic regression” in the original version was used to imply the logistic regression with a single predictor. We have rephrased the description of the analytic method to make it clear.

4. The outcome variable (NSSI at 6 month) is quantitative-how was then “NSSI initiation or cessation defined? How was the frequency of it used- was a proportional odds logistic regression model used? If so, this should be described in the Methods section and again mentioned in the results section.

Response: Thank you. Participants’ NSSI were reported on a 5-point scaleresponses were then recoded to a binary categorical variable. That is, participants who reported 1 “never” for all the nine self-injury items were coded as 0 (no NSSI); participants who reported 2 “seldom” to 5 “always” for any self-injury item were coded as 1 (have NSSI). “NSSI initiation” was defined as no NSSI at baseline but have NSSI at follow-up, “NSSI cessation” was defined as have NSSI at baseline but no NSSI at follow-up. In our analyses, binary logistic regression models are used for binary categorical dependent variables (i.e., recoded NSSI responses). Thus, the proportional odds logistic regression model was not used in our analyses. According to your advice, we have added more information to describe the analytic method clearly.

5. Were the models single predictor, adjusted only for gender at SE? What does the “adjusted analyses” referred to in line 211 mean?

Response: Thank you. The bivariate models were single predictors, adjusted only for gender or SES. We have replaced the term “adjusted analyses” with “bivariate logistic regression models” and added note in Tables to make it easier to understand.

6. Language mistakes are on mild to moderate level- stopped correcting them after line 130. *Grammar mistake: Line 100, “Four junior high schools agree” should be “Four junior high schools agreed” *Extra comma on line 118: “Besides, a wide range of interpersonal and intrapersonal factors” *Line 123: “studying variables” should be “study variables, “and had been successfully used” should be “that had been successfully used”

Response: Following your suggestion, we have revised the sentence in line 100, line 118, and line 123 respectively. Besides, we have carefully revised the full manuscript to correct language mistakes.

Round 2

Reviewer 1 Report

In this second version of the manuscript authors made a great effort to improve it following reviewers’ recommendations. I thank authors for the job. To my view, authors addressed reviewers' comments satisfactorily. I just recommend authors to pay attention to several English grammar mistakes that can be found throughout the text in a careful re-editing. 

Author Response

In this second version of the manuscript authors made a great effort to improve it following reviewers’ recommendations. I thank authors for the job. To my view, authors addressed reviewers’ comments satisfactorily. I just recommend authors to pay attention to several English grammar mistakes that can be found throughout the text in a careful re-editing.

Response: Thank you very much for your suggestion. Following your suggestion, we have polished the entire manuscript using a language editing service provided by MDPI Author Services before resubmission.

Reviewer 2 Report

The authors addressed my points, however there has been a change of analysis that may raise a new issue. Overall, the single-predictor models’ analysis has improved, as it now includes multiple testing adjustment of the significance levels, using the Benjamini-Hochberg method, which resulted in a change in conclusions- no predictors were found for NSSI cessation. Please change the terminology for the B-H adjustment- it is a correction for multiple testing, not multiple comparison, which would only be applicable for multi-group studies.

The new multi-predictor logistic regression described in Table 3 has   many predictors and may be overfitting the data. The rule of thumb is that there should be 10-15 “cases” or “controls”, whichever is smaller, for each predictor. This condition is not violated. But  some of these predictors are likely highly correlated, which can mean that they explain away each other’s predictive power, resulting in few or no significant effects. That is the case in the present study. It is not clear what including so many variables in the same model will achieve. If the aim was to arrive at a parsimonious model, a LASSO model selection strategy could be used. If the goal was to adjust intrapersonal and interpersonal factors for each other’s effects, then a dimension reduction strategy like factor analysis should be applied before the logistic regression model is fit.

Also, multivariate logistic regression is better described as “multipredictor”- multivariate models refer to models with multiple outcome variables, while multipredictor models to those with multiple predictor variables.

Further language or grammar mistakes:

  1. Methods section, line 224-225: “demographic (i.e., gender, age, left-behind, and subjective

225 SES) and descriptive (percentage, mean, and SD) characteristics of the study variables” - the last word surely should be “sample” or “subjects”? Variables do not have demographic characteristics, people (and groups of people) do.

  1. Line 247-“ should be No significant differences… were found” (not “was found”)

Author Response

The authors addressed my points, however there has been a change of analysis that may raise a new issue. Overall, the single-predictor models’ analysis has improved, as it now includes multiple testing adjustment of the significance levels, using the Benjamini-Hochberg method, which resulted in a change in conclusions- no predictors were found for NSSI cessation. Please change the terminology for the B-H adjustment- it is a correction for multiple testing, not multiple comparison, which would only be applicable for multi-group studies.

Response: Thank you very much for your correction. Following your advice, we have changed the terminology.

The new multi-predictor logistic regression described in Table 3 has many predictors and may be overfitting the data. The rule of thumb is that there should be 10-15 “cases” or “controls”, whichever is smaller, for each predictor. This condition is not violated. But some of these predictors are likely highly correlated, which can mean that they explain away each other’s predictive power, resulting in few or no significant effects. That is the case in the present study. It is not clear what including so many variables in the same model will achieve. If the aim was to arrive at a parsimonious model, a LASSO model selection strategy could be used. If the goal was to adjust intrapersonal and interpersonal factors for each other’s effects, then a dimension reduction strategy like factor analysis should be applied before the logistic regression model is fit.

Response: Thank you very much for your suggestion. We explored a statistical model on how resulting relevant variables in the previous bivariate analyses relate to predict self-injury and try to determine the independent contribution of each variable in the model, which is suggested by Review 1. Following your advice, we have added information of our study aim in Data Analysis section, which is highlighted in yellow, to make it clear.

Also, multivariate logistic regression is better described as “multipredictor”- multivariate models refer to models with multiple outcome variables, while multipredictor models to those with multiple predictor variables.

Response: Thank you. Following your advice, we have described the model as “multi-predictor” model.

Further language or grammar mistakes:

*Methods section, line 224-225: “demographic (i.e., gender, age, left-behind, and subjective SES) and descriptive (percentage, mean, and SD) characteristics of the study variables” - the last word surely should be “sample” or “subjects”? Variables do not have demographic characteristics, people (and groups of people) do.

*Line 247-“should be No significant differences… were found” (not “was found”)

Response: We apologize for the language and grammar mistakes. Following your suggestion, we have revised the sentence respectively. Besides, we have polished the entire manuscript using a language editing service provided by MDPI Author Services before resubmission.